# A Human Stem Cell-Derived Brain-Liver Chip for Assessing Blood-Brain-Barrier Permeation of Pharmaceutical Drugs

**DOI:** 10.3390/cells11203295

**Published:** 2022-10-19

**Authors:** Leopold Koenig, Anja Patricia Ramme, Daniel Faust, Manuela Mayer, Tobias Flötke, Anna Gerhartl, Andreas Brachner, Winfried Neuhaus, Antje Appelt-Menzel, Marco Metzger, Uwe Marx, Eva-Maria Dehne

**Affiliations:** 1TissUse GmbH, Oudenarder Str. 16, 13347 Berlin, Germany; 2Pharmacelsus GmbH, Science Park 2, 66123 Saarbrücken, Germany; 3Competence Unit Molecular Diagnostics, Austrian Institute of Technology GmbH, Giefinggasse 4, 1210 Vienna, Austria; 4Department of Medicine, Danube Private University, Steiner Landstraße 124, 3500 Krems an der Donau, Austria; 5Chair of Tissue Engineering and Regenerative Medicine, University Hospital Würzburg, Röntgenring 11, 97070 Würzburg, Germany; 6Translational Center for Regenerative Therapies, Fraunhofer Institute for Silicate Research, Röntgenring 11, 97070 Würzburg, Germany; 7Department of Medical Biotechnology, Institute of Biotechnology, Technische Universität Berlin, Gustav-Meyer-Allee 25, 13355 Berlin, Germany

**Keywords:** blood-brain barrier (BBB) model, human induced pluripotent stem cells (hiPSCs), microphysiological systems (MPS), multi-organ chip, brain–liver chip

## Abstract

Significant advancements in the field of preclinical in vitro blood-brain barrier (BBB) models have been achieved in recent years, by developing monolayer-based culture systems towards complex multi-cellular assays. The coupling of those models with other relevant organoid systems to integrate the investigation of blood-brain barrier permeation in the larger picture of drug distribution and metabolization is still missing. Here, we report for the first time the combination of a human induced pluripotent stem cell (hiPSC)-derived blood-brain barrier model with a cortical brain and a liver spheroid model from the same donor in a closed microfluidic system (MPS). The two model compounds atenolol and propranolol were used to measure permeation at the blood–brain barrier and to assess metabolization. Both substances showed an in vivo-like permeation behavior and were metabolized in vitro. Therefore, the novel multi-organ system enabled not only the measurement of parent compound concentrations but also of metabolite distribution at the blood-brain barrier.

## 1. Introduction

The blood-brain barrier (BBB) is one of the tightest barriers in the human body. It has the delicate task to handle nutrient supply, waste removal, and protection from harmful substances and pathogens, thereby enabling homeostasis of the human brain. The BBB is formed by specialized brain microvascular endothelial cells (BMECs), but multiple other cell types such as astrocytes and pericytes are known to be involved in the maintenance and control of the barrier function. 

A promising tool for the generation of an unlimited BMEC supply has emerged in the last decade by the targeted differentiation of BMEC-like cells from embryonal or human induced pluripotent stem cells (hiPSC) [1] and culture on cell culture inserts. The direct translation of iPSC-derived BMEC-like cells and primary BMEC-based cell culture insert models towards microphysiological systems (MPS) resulted in the development of two-dimensional microfluidic models. These models usually consist of two or more channels, which are separated by a porous membrane sandwiched in between PDMS layers. Endothelial cells and secondary cell types such as astrocytes, pericytes and neurons can be cultured on opposite sides of the membrane. The continuous perfusion of the models is usually generated by external syringe pumps [2,3,4] or external peristaltic pumps [5,6,7]. In most devices, secondary cell types grow in a monolayer on the membrane or on the channel walls, but some designs have incorporated hydrogels on the abluminal side of the membrane allowing the three-dimensional arrangement of secondary cell types such as astrocytes and pericytes [3,4]. The next logical step in the development of advanced microphysiological organ models is the combination of those models in complex multi-organ chips.

It has been shown previously that the combination of multiple organ models in a combined microfluidic circuit enhances organ functionality and promotes maturation [8,9]. Still, only a few publications describe the interconnection of a BBB model with additional tissue models. In 2017, the functional coupling of an intestinal, liver, kidney and blood–brain barrier model was published. Although, in this study, the organ models were not physically coupled but efflux medium was transferred between multiple MPSs, which were distributed in four different laboratories [10]. Notably, the metabolization of trimethylamine and vitamin D3 by the liver module and the subsequent BBB permeation of the metabolites was shown. In 2020, the successful coupling of eight individual organ chips incorporating the intestine, liver, kidney, heart, lung, skin, blood-brain barrier and brain was published [11]. In this publication, the BBB consists of primary BMECs, human astrocytes and human brain pericytes, as described in a previous publication of the same group [5]. Similarly to the other approaches, a mixed donor background was used, and organ communication was allowed solely by transferring medium supernatants between individual chips. The advantages of a direct organ model interconnection such as the measurement of liver-induced metabolite permeation or changes in BBB permeation due to kidney excretion have not been explored so far in this setup. In a recent publication, a liver spheroid model consisting of HepG2 cells and a glioblastoma model were separated from each other in an MPS by a BBB model composed of BMECs [12]. Three marketed antitumor drugs: paclitaxel, capecitabine and temozolomide were tested in the system, and the BBB model-dependent changes in the observed effects of the compounds were shown. This study showed the value of the interconnection of a liver and a BBB/brain model by enabling the analysis of liver metabolization, BBB permeation and tumor clearance efficacy at the same time.

Here, we describe for the first time the simultaneous cultivation of a fully autologous iPSC-derived human blood-brain barrier, cortical brain and liver model in a closed microfluidic circuit. The testing of two model compounds revealed in vivo-like pharmacokinetic behavior. Therewith, the here-presented MPS has multiple advantages over classical cell culture inserts and animal models. Compared to cell culture insert models, the MPS enables the application of fluid flow-induced shear stress and interconnection with additional organ models. Compared to animal models, the MPS uses human cells to allow better emulation of human pharmacokinetics and dynamics.

## 2. Materials and Methods

### 2.1. Cultivation and Maintenance of Human-Induced Pluripotent Stem Cells

All differentiated tissues used for the organ chip coculture were derived from the same hiPSC line HUMIMIC101 (TissUse GmbH, Berlin, Germany). HUMIMIC101 cells were generated by TissUse GmbH with donor consent for commercial use to study the influence of age and gender on the reprogramming into iPSCs and differentiation into specific cell types [13]. The cells were maintained on growth factor reduced (GFR) Matrigel coated 6-well plates (Corning, Corning, NY, USA) in StemMACS iPS-Brew XF (Miltenyi Biotec, Bergisch Gladbach, Germany). During the normal maintenance culture, the cells were seeded at densities in the range of 2000–4000 cells/cm² in StemMACS iPS-Brew XF with the addition of 10 µM of ROCK-inhibitor Y-27632 (Cayman Chemicals, Ann Arbor, MI, USA). After 48 h, the medium was removed and replaced by fresh medium without the ROCK-inhibitor. From there on, the media were exchanged every day until the passaging of the cells seven days after seeding when the confluence reached approximately 90%. For passaging, hiPSCs were washed once with PBS without calcium and magnesium and incubated with a sufficient volume of Accutase (Corning, Corning, NY, USA) for five to seven minutes. 

### 2.2. Differentiation of BMEC-like Cells

The differentiation into BMEC-like cells was carried out as described by others [1,14]. HiPSCs with a density of 25,000 to 35,000 cells/cm² were treated for 6 days with BBB stage 1 medium (DMEM/F-12 + 20% KnockOut Serum Replacement + 0.1 mM β-Mercaptoethanol (all Thermo Fisher Scientific, Waltham, MA, USA) + 1% MEM NEAA + 1 mM L-Glutamine (both Corning, Corning, NY, USA)) for endothelial and neural cell co-differentiation. After that, the medium was exchanged to BBB stage 2 medium (Endothelial-SFM (Thermo Fisher Scientific, Waltham, MA, USA) + 1% platelet-poor plasma-derived bovine serum (Alfa Aesar, Haverhill, MA, USA)) supplemented with 10 µM retinoic acid (Sigma-Aldrich, St. Louis, MO, USA) and 20 ng/mL FGF-2 (154 a.a.) (Peprotech, Cranbury, NJ, USA) for two days. After 2 days, cells were dissociated with Accutase and seeded with 1 × 10^6^ cells/cm² on the bottom side of 96-well 0.4 µm pore size polyester membrane Transwells (Corning, Corning, NY, USA). Transwell membranes were coated before with 400 mg/mL collagen IV (Sigma-Aldrich) and 100 mg/mL fibronectin (Thermo Fisher Scientific, Waltham, MA, USA) in cell culture grade water for 4 h at 37 °C. Cells were allowed to attach to the membranes for 2.5 h before the Transwells were flipped upside down and cultured in a 96-well Transwell receiver plate (Corning, Corning, NY, USA) for 1 day in BBB stage 2 medium supplemented with 10 µM retinoic acid and 20 ng/mL FGF-2. One day after seeding, the medium was exchanged for BBB stage 2 medium without supplements.

### 2.3. Differentiation of Neural Spheroids

Differentiation into neural spheroids was carried out by a newly developed bioreactor cultivation protocol based previously published protocols describing the differentiation of iPSC-spheroids into cortical neurospheres [15,16]. Differentiation was carried out as described in the Eppendorf application note 364 [17]. After complete differentiation in the bioreactor, spheroids were resuspended in Bambanker cryopreservation solution (Nippon Genetics, Tokyo, Japan) and cryopreserved until further usage. For the thawing of neural spheroids, cryovials were removed from the −180 °C storage and thawed gently in the water bath for 90 s. Then, the spheroid suspension was transferred from the cryovial into a 50 mL centrifugation tube filled with 20 mL of neural thawing medium (DMEM/F12 (Thermo Fisher Scientific, Waltham, MA, USA) + 10% fetal calf serum (Corning, Corning, NY, USA)). Spheroids were washed with the neural thawing medium twice and resuspended in neural cultivation medium (Neurobasal medium + 1% N2 supplement, 2% B27 supplement without vitamin A (all Thermo Fisher Scientific, Waltham, MA, USA), 2 mM L-glutamine, 1% MEM nonessential amino acids, 1% penicillin-streptomycin (all Corning, Corning, NY, USA), 20 ng/mL BDNF and 20 ng/mL GDNF (both Peprotech, Cranbury, NJ, USA)). Neural spheroids were cultured in 125 mL baffled Erlenmeyer flasks (Corning, Corning, NY, USA) at an orbital shaker with 85 rpm in the incubator for at least two weeks before they were used for coculture experiments. 

### 2.4. Assembly of BBB/Brain Models

Neural spheroids with a total amount of 0.8 to 1.2 × 10^6^ cells were transferred into the BMEC-like cell-seeded Transwell, 24 h after the seeding of the BMEC-like cells. From there, the respective culture medium of the BMEC-like cells was used for the cultivation of the neural spheroids in the BBB model.

### 2.5. Differentiation of Mesenchymal Stromal Cells

HUMIMIC101-derived mesenchymal stromal cells (MSC) were differentiated by an adapted protocol [18] as described previously [9] and cryopreserved in passage 2. The cryopreserved MSCs were thawed briefly at 37 °C in the water bath and then seeded with 10,000 cells/cm² in mesenchymal culture medium (low glucose DMEM with 10% FCS, 1% penicillin/streptomycin (all Corning, Corning, NY, USA) and 10 ng/mL FGF-2 (146 a.a.) (Peprotech, Cranbury, NJ, USA)). The cells were used for liver spheroid formation below passage 6.

### 2.6. Differentiation of Endothelial Cells

Differentiation into endothelial cells was carried out based on an adapted protocol [19]. HiPSCs were initially seeded with a density of 1500–2000 cells/cm² and switched to STEMdiff APEL 2 medium (Stemcell Technologies, Vancouver, BC, Canada) supplemented with 6 µM CHIR99021 (Sigma-Aldrich, St. Louis, MO, USA) after three days. After two days this medium was exchanged for STEMdiff APEL 2 medium supplemented with 10 ng/mL FGF-2 (146 a.a.) (Peprotech, Cranbury, NJ, USA), 25 ng/mL BMP4 (Miltenyi Biotec, Bergisch Gladbach, Germany) and 50 ng/mL VEGF (Peprotech, Cranbury, NJ, USA). After two days, the cells were passaged with Accutase and reseeded with 10,000 cells/cm² in endothelial cell growth medium MV2 (PromoCell, Heidelberg, Germany) supplemented with 50 ng/mL VEGF and 2 µM forskolin (Cayman Chemical, Ann Arbor, MI, USA). From there on, the cells were split every four days with Acctuase with media exchanges every other day. At day 12 of the differentiation, the cells were used for the formation of liver spheroids. 

### 2.7. Differentiation of Hepatocyte-like Cells

Differentiation into hepatocyte-like cells (HLCs) was carried out based on an adapted protocol [20]. hiPSCs were seeded with a density of 33,000 cells/cm² on 0.5 μg/cm² rLaminin521 (Corning, Corning, NY, USA). The iPSC culture medium was supplemented with STEMdiff DE TeSR-E8 supplement (Stemcell Technologies, Vancouver, BC, Canada). Two days after the seeding of the iPSC, the differentiation was initiated by exchanging the medium with STEMdiff DE Basal Medium including the provided supplements MR and CJ (Stemcell Technologies, Vancouver, BC, Canada). For the next three days, the medium was exchanged daily with STEMdiff DE Basal Medium including only the supplement CJ. On day 4 after the differentiation start, the medium was then exchanged for KnockOut DMEM supplemented with 5% KnockOut serum replacement (both Thermo Fisher Scientific, Waltham, MA, USA), 1 mM L-glutamine, 1% MEM nonessential amino acids (both Corning), 0.1 mM beta-mercaptoethanol, 20 ng/mL BMP4, 20 ng/mL FGF-4 (both Miltenyi Biotec, Bergisch Gladbach, Germany) and 1× penicillin-streptomycin. The medium was refreshed every other day until day 10 of the differentiation protocol. On day 10 of the differentiation, the medium was exchanged for liver maturation medium (HepatoZYME-SFM supplemented with 2% KnockOut serum replacement (both Thermo Fisher Scientific, Waltham, MA, USA), 2 mM L-glutamine, 1% MEM nonessential amino acids, 1% penicillin-streptomycin, 10 ng/mL HGF, 10 ng/mL oncostatin M (both Miltenyi Biotec, Bergisch Gladbach, Germany) and 0.1 μM dexamethasone (Sigma Aldrich, St. Louis, MO, USA)). The medium was refreshed every other day until day 15 of the differentiation protocol. At this timepoint, HLCs were used for liver spheroid formation.

### 2.8. Assembly of Liver Spheroids

hiPSC-derived hepatocyte-like cells (HLCs) were detached by incubation with 0.25% Trypsin/2.21 mM EDTA (Corning, Corning, NY, USA) for 15 min in the incubator. Detached cells were collected in a 50 mL centrifugation tube and the flasks were washed with liver spheroid medium (liver maturation medium supplemented with 2% fatty acid-free bovine serum albumin (Sigma Aldrich, St. Louis, MO, USA) and 10 μM Rock inhibitor Y-27632 to collect remaining cells. The cell suspension was then filtered through a 70 μm mesh size cell strainer (Corning, Corning, NY, USA) to remove the extracellular matrix and cell aggregates. hiPSC-derived endothelial cells were detached by incubation with Accutase for 7 min and hiPSC-derived mesenchymal stromal cells (MSCs) were detached by incubation with 0.05% Trypsin/0.53 mM EDTA for 5 min in the incubator. Detached cells were collected in a 50 mL centrifugation tube and the dishes were washed with liver spheroid medium to collect the remaining cells. All cell suspensions were centrifuged, the supernatant was discarded, the cell pellet was resuspended in liver spheroid medium and counted. The final cell suspension was mixed with a ratio of 80% HLCs, 15% endothelial cells, and 5% MSCs at a concentration of 666,667 cells/mL in liver spheroid medium. The cell mixture was seeded into 384-well U-bottom ULA plates (Nexcelom Bioscience, Lawrence, MA, USA) with 75 µL per well, which corresponds to a cell number of 50,000 cells per well. The 384-well plate was centrifuged for 2 min at 100× *g* and then incubated for 72 h on a multi-function 3D rotator (Grant Instruments, Shepreth, UK). The rotator was programmed to switch constantly between 15 s of orbital rotation at a 40° angle and 5 s of vibration at a 5° angle. After 72 h, spheroids were collected and pooled with 20 spheroids per well of a 24-well flat-bottom ULA plate (Corning, Corning, NY, USA). The medium was then exchanged with liver maturation medium. At this point, the spheroids were ready for transfer into the Chip4.

### 2.9. Multi-Organ Chip Experiments

On the day of the experiment start, the Chip4 (TissUse GmbH, Berlin, Germany) was filled with Williams E medium (PAN-Biotech, Aidenbach, Germany) plus 5% human AB serum, 1% non-essential amino acids, 2 mM L-glutamine, 1% penicillin-streptomycin and 1% ITS solution (all Corning, Corning, NY, USA), referred to as chip coculture medium supplemented with 10 μM ROCK-inhibitor Y-27632. Each compartment was opened and filled separately, starting with 300 μL in the mix 1 compartment, followed by 1000 μL in the intestinal compartment, 300 µL in the liver compartment and 150 μL in the brain compartment.

To insert the liver model, 20 liver spheroids were pipetted into the liver compartment. Cut-off 96-well Transwell models with neural spheroids with or without BMEC-like cells at the bottom side of the membrane were filled with 75 µL chip coculture medium and transferred with tweezers into the brain compartment. 

The chips were connected to a HUMIMIC starter (TissUse GmbH, Berlin, Germany) and cultured in the incubator with the pump set to a frequency of 1 Hz and pressure/vacuum set to +/− 700 mbar.

The first medium exchange was performed two days after the assay was initiated. First, the mix 1 compartment was opened, and a full medium exchange with 300 μL chip coculture medium was performed. Then, the intestinal compartment was opened, and a full medium exchange with 1000 μL chip coculture medium was performed. At day five, atenolol and propranolol were added to the application medium. Only 1300 μL of the complete chip media volume of 1782 μL was replaced during the medium exchange. Therefore, the substance concentration in the application medium was concentrated 1.37-fold higher than the targeted concentration of 10 μM in the Chip4 (Appendix A). As a solvent control, an equal amount of DMSO was added to the control chip medium. On day five, the medium in the brain compartment was renewed with 75 μL chip coculture medium. After that, the medium exchange was performed as described for day two, with substance application medium in the treatment group and DMSO solvent control medium in the control group.

At day 7, 48 h after substance application, the Transwell was removed from the brain compartment with tweezers. The medium inside the Transwell was sampled for LC/MS analysis. The Transwell was then filled again with 75 μL chip coculture medium and was used for TEER measurement with the EVOM2 instrument and an STX100C96 electrode (both World Precision Instruments, Sarasota, FL, USA). Then, the liver compartment was opened, and the medium was sampled for LC/MS analysis. Liver spheroids were removed from the compartment and transferred into a prepared sample tube. BMECs on membranes, neural spheroids and liver spheroids were either fixated with 4% PFA and used for immunofluorescence staining or used for RNA extraction with the NucleoSpin RNA XS Kit (Machery-Nagel, Düren, Germany) for subsequent transcript quantification. Detailed information on immunofluorescence staining procedure and qPCR procedures is given in the Appendix A. Details of the used primers for qPCR are listed in Appendix A and details of the used antibodies for immunocytochemistry are listed in Appendix A.

Medium supernatants that were not used for the LC/MS analysis of substance concentrations were used for measurement of the glucose, lactate, albumin and lactate dehydrogenase (LDH) concentration with the Indiko clinical chemistry analyzer (Thermo Fisher Scientific, Waltham, MA, USA) with commercially available measurement kits (Appendix A). Detailed information on substance quantification by LC/MS and Indiko clinical chemistry analyzer is given in the Appendix A.

### 2.10. Statistics

The GraphPad Prism software (GraphPad Software, San Diego, CA, USA) was used to plot the data and perform a comparative analysis between the means of different conditions. For all statistical comparisons, an alpha value of 0.05 was used. All comparative analysis on gene expression data was performed on logarithmic 2^−ΔCt^ values because of the lognormal distribution of gene expression data.

For comparing the means of liver gene expression with and without chip culture, a one-way ANOVA was performed using the Dunnett post hoc test to correct for multiple comparisons. The day 0 static sample group was used as the control group. 

For comparing the means of the TEER values before and after chip culture and with and without atenolol and propranolol application, a one-way ANOVA was performed using the Tukey post hoc test to correct for multiple comparisons.

To compare the gene expression data of multiple genes between two test groups acquired by the multiplex BBB qPCR chip, the means were compared using multiple *t*-tests. Statistical significance was determined using the Holm–Sidak method to correct for multiple comparisons. Each gene was analyzed individually, without assuming a consistent standard deviation (SD). 

For comparing two unpaired means of atenolol and propranolol concentrations and their metabolites between the medium circulation and the brain compartment, a two-tailed Student’s *t*-test was performed. 

For comparing the means of the concentration data of glucose, lactate, LDH and albumin, a two-way ANOVA was performed using the Tukey post hoc test to correct for multiple comparisons.

## 3. Results

### 3.1. Setup of the Coculture Assay

Neural spheroids were differentiated from the iPSC line HUMIMIC101 [17] and combined in a 96-well Transwell model with brain microvascular endothelial (BMEC)-like cells [1] derived from the same donor. During the differentiation process of the neural spheroids in the bioreactor and the Erlenmeyer flask, spheroids increased the transcript expression of the cortical developmental marker TBR1, the glutamatergic marker SLC17A6 and the GABAergic marker GAD1 (Appendix A), indicating that they followed their intended differentiation route towards a dorsal forebrain identity [15,16].

In parallel, human iPSC-derived liver spheroid models were formed from hepatocyte-like cells [20], endothelial cells and stromal cells. After three days of spheroid formation, 20 of these spheroids constituted one liver model (Figure 1A). To test the metabolic potential, the generated spheroids were treated with 5 μM propranolol for three days. The formation of metabolites was measured by LC/MS, 24 h and 72 h after propranolol was first applied. For propranolol, two metabolites, an oxidized metabolite M1 and a glucuronidated metabolite M2, were identified in the medium supernatant (Appendix A). In addition, CYP3A4 activity was induced compared to the DMSO solvent control by stimulation with rifampicin, with higher inducibility upon longer culture times (Appendix A). For one differentiation, the transcript levels of multiple CYP enzymes were measured and CYP3A4 activity was found to be upregulated significantly about threefold in the rifampicin group compared to the untreated control and the DMSO solvent control (Appendix A).

This liver model was subsequently transferred into the HUMIMIC Chip4 [9] and combined with the BBB/brain model. In this study, the PDMS layer 1 that normally contains the microfluidic circulation for the kidney model was replaced by a PDMS layer without a microfluidic circuit (Figure 1B). The BBB/brain Transwell model inserted into the brain compartment generated a 100 µm high channel underneath so that the BMEC-like cells cultured on the bottom side of the Transwell membrane were directly exposed to the fluid flow (Figure 1C).

After an initial five-day adjustment phase of the coculture, propranolol and atenolol were applied through the intestinal and mix 1 compartment (Figure 2A). Over the total culture time of seven days, glucose consumption and lactate production stabilized at 1.20 (SD ± 0.14) µmol and 1.22 (SD ± 0.25) µmol per day (Figure 2B). The stable consumption of glucose and production of lactate is an indicator for constant tissue homeostasis in the Chip4 without excessive proliferation or cell death changing the overall cell number. In three experiments, the lactate/glucose ratio was close to one after day 2 (Figure 2B). A ratio of one indicates that approximately 50% of the used glucose is used for energy generation by oxidative phosphorylation. This rate is an indicator for the differentiation status and maturity compared to hiPSCs, which are known to receive the vast majority of their energy supply from hydrolysis [21]. 

### 3.2. Organ Models in the HUMIMIC Chip4-Liver Model

Liver models cultured in the Chip4 attached slightly to the compartment bottom but maintained their spheroidal shape over the whole culture time (Figure 3A). Spheroids cultured in the Chip4 over 7 days were compared to spheroids before Chip4 culture and to spheroids that were cultured in liver maturation medium under static conditions for the same timespan by the analysis of characteristic gene transcript marker expression (Figure 3B). Alpha-fetoprotein (AFP) expression was increased non-significantly during the chip culture and was expressed significantly higher compared to static-cultured spheroids. The expression of the major transport protein albumin increased significantly over time, irrespective of the culture conditions. The pronounced expression of albumin was also detected before and after the chip culture by immunofluorescence staining (Figure 3C). 

The cytochrome P450 member CYP3A4 was upregulated significantly over time both under static as well as under dynamic conditions with a slightly higher expression under dynamic conditions, while the expression of the multidrug resistance-associated protein 2 (MRP2) remained constant in both culture conditions (Figure 3B). The expression of the nuclear transcription factors FOXA2 and HNF4a decreased over time in the static culture setup to a bigger extent than in dynamic HUMIMIC Chip4 cultivation. Stability of the housekeeper gene in iPSC and differentiated liver models were shown by stable cT values over the different conditions (Appendix A). The expression of the nuclear transcription factors HNF4a was also confirmed before and after chip culture by immunofluorescence staining (Figure 3C).

### 3.3. Organ Models in the HUMIMIC Chip4-BBB/Brain Model

BMECs on the Transwell membranes kept their monolayer morphology with no visible cell aggregates (Figure 4A). Sodium-fluorescein permeation before chip insertion was 0.45 ± 0.27 µm/min in the first and 0.1 ± 0.04 µm/min in the second experiment. The procedure of sodium-fluorescein permeation measurement is given in the Appendix A. The TEER values differed between individual Transwells and ranged from 469 to 1467 Ohm·cm² before chip insertion in two experiments. The average TEER values decreased over time and ranged from 101 to 1160 Ohm·cm² after 7 days of culture. One model, where TEER values dropped below 150 Ohm·cm², was excluded from analysis at day 7 of the assay. This reduced barrier tightness was observed to be independent from the application of propranolol and atenolol (Figure 4B). 

Neural spheroids flattened and grew out over time (Figure 4A). They showed high numbers of proliferating cells in the border regions of the spheroid up to a depth of approximately 200 µm, and apoptotic cells appeared in the core region of the spheroid (Figure 4F). Nevertheless, the tissue structure was maintained, and βIII-tubulin (TUBb3) positive neuronal cells throughout the model and glial fibrillary acidic protein (GFAP) positive glial cells close to the spheroid surface were still present at the end after one week of Chip4 culture (Figure 4E). In addition to the rather early neuronal marker TUBb3, the more mature neuronal marker MAP2 and the synaptic marker synaptophysin were also expressed (Figure 4G).

The analysis of 90 BBB-related genes on transcript levels showed marked differences between BMECs before and after Chip4 culture, but not between atenolol/propranolol and non-treated chips (Appendix A). Therefore, a comparison of the mean expression levels was carried out between the day 0 and day 7 samples independent of the treatment background. Among the upregulated genes, four tight-junction-associated proteins were discovered: the three claudins 1, 8 and 9 and the junctional adhesion molecule 2 (JAM2). Furthermore, the glucose transporter GLUT1, which is the main glucose transporter at the human BBB and AQP3 was upregulated. All of the discovered upregulated genes were also upregulated in a sample of static cultured BMEC-like cells with neural spheroids (Figure 4D). Stability of the two selected housekeeper genes in iPSC and differentiated BMECs were shown by stable cT values over the different conditions (Appendix A).

In the group of downregulated genes, the two tight-junction-associated proteins claudin-10 and 11 were discovered. Two genes associated with the cytoskeleton: cytokeratin 18 and the S100 calcium-binding protein A4 (S100A4) were also among the downregulated genes. The downregulation of cytoskeleton-associated genes was likely related to a reduced need to restructure the cytoskeletal organization once a tight cellular network had formed. The cell surface glycoprotein MUC18, also called the melanoma cell adhesion molecule, is expressed by endothelial cells and was downregulated at day 7 of Chip4 culture compared to the starting point. Cytokeratin 18 and MUC18 were only downregulated in the dynamic cultivation but not in a sample of BMEC-like cells cultured under static conditions, indicating that these changes are associated with the dynamic Chip4 culture (Figure 4D).

BBB marker genes, ZO-1, claudin-5, GLUT1 and VE-cadherin, were expressed before and after the Chip4 culture (Figure 4C). The expression of the junction marker VE-cadherin, claudin-5 and ZO-1 appeared more uniformly associated with the cell–cell junctions over time, with no apparent changes in cell orientation, considering the cytoskeleton staining with F-actin (Figure 4C).

### 3.4. Permeation and Metabolization of Atenolol and Propranolol

Glucose, lactate, LDH and albumin concentrations were analyzed in the medium circulation and the brain compartment before the addition of atenolol and propranolol at day 5 of the Chip4 culture. The glucose concentration was significantly lower in the brain compartment compared to the medium circulation, independent of the presence of BMEC-like cells (Figure 5A). Matching the lower glucose concentrations in the brain compartment, the lactate concentration was significantly higher in the brain compartment compared to the medium circulation independent of the presence of BMEC-like cells (Figure 5A). LDH and albumin are, compared to glucose and lactate, large molecules, with molecular weights of 144 kDa and 67 KDa, respectively, that in vivo are restricted from BBB passage. In the Chip4, the albumin and LDH concentrations were balanced between the brain compartment and circulation without the BMEC-like cells, but a gradient was formed when BMEC-like cells were present (Figure 5A). Higher LDH values are likely to be a result of the low BBB permeability of LDH, preventing LDH formed in the brain compartment to enter the medium circulation. The albumin gradient shows that the BMEC-like cells act as a functional barrier for albumin in the Chip4. The measured albumin is most likely the remaining albumin from the experiment start when the compartment was filled with chip coculture medium containing albumin due to the contained human serum.

After 5 days of coculture in the Chip4, atenolol and propranolol were applied for 48 h through the intestinal and mix 1 compartment. The binding behavior of the two substances was measured in cell-free control chips over 48 h. Pronounced binding was measured for propranolol with a mean recovery rate of 38.4% (4.4% SD). Neglectable binding was measured for atenolol with a mean recovery rate of 97.4% (16.6% SD). A mean concentration of propranolol of 3.40 µM in the medium circulation and 4.86 µM in the brain compartment was measured in the presence of BMEC-like cells. For atenolol, a mean concentration of 7.55 µM in the medium circulation and 2.13 µM in the brain compartment was measured (Figure 5B). This translates into mean brain/circulation concentration ratios of 1.46 for propranolol and 0.28 for atenolol in the presence of BMEC-like cells, while both ratios were close to 1 without BMEC-like cells (Figure 5C).

Metabolite formation was identified by LC/MS based on accurate mass. For atenolol, a desaturated metabolite (M1), a dealkylated and oxidized metabolite (M2) and a dealkylated and glucuronidated metabolite (M3) were identified in the media supernatants (Figure 5D). 

Dealkylated atenolol and atenolol glucuronide were found in higher concentrations in the brain compartment (Figure 5E), indicating that it either was formed there by the cells of the neural model or was actively transported by the BMEC-like cells. Atenolol glucuronide is formed by UGT1A6, which is expressed in the brain by astrocytes [22]. For the neural spheroids, we found a correlation between the expression of the astrocyte marker GFAP and the UDP-glucuronosyltransferase 1A4 and 1A6 on transcript level (Appendix A). GFAP was expressed in the neural spheroids, which were 80 days old for this experiment, as seen in immunofluorescence staining (Figure 4G). One explanation for the observed distribution pattern is that the glucuronidation was facilitated by astrocytes in the neural spheroids, and the distribution of the metabolite into the medium circulation was prevented by the BMEC-like cell barrier. 

For propranolol, an oxidized metabolite (M1) and a glucuronidated metabolite were identified in the media supernatants (Figure 5D). Oxidized propranolol was detected at higher concentrations in the application medium and cell-free control Chip4 and was therefore not specifically metabolized by the liver model (Figure 5E). This is likely due to the low expression of CYP2D6 in the liver model (Appendix A), which is the responsible enzyme for the oxidation of propranolol [23]. 

The glucuronidated propranolol metabolite M2 was not detected in the cell-free Chip4, indicating a cell specific metabolization. Without BMEC-like cells, the metabolite M2 was equally distributed between the circulation and the brain compartment. With BMEC-like cells, the metabolite M2 was restricted from BBB passage as seen by a significant difference in the concentration between medium circulation and brain compartment (Figure 5E). 

## 4. Discussion

The simultaneous cultivation of an autologous iPSC-derived liver and BBB/brain model was achieved in the microfluidic HUMIMIC Chip4 cultivation system. hiPSC-derived liver spheroids maintained their differentiation status in the Chip4. While increasing expression of CYP3A4 and albumin pointed in the direction of ongoing maturation, increasing the expression of alpha-fetoprotein rather indicated a fetal phenotype. However, even the most recent protocols for the generation of hepatocyte-like cells from hiPSC showed that the phenotype of these cells resembles rather a fetal than an adult liver [24]. Consequently, a fetal phenotype has to be expected, and the upregulation of alpha-fetoprotein accompanied by an upregulation of albumin can still be interpreted as an indicator of maturation. In addition, it has to be kept in mind, that the chip coculture medium did not contain any additionally added growth factors on top of the growth factor background that comes with the human serum.

Others have shown that perfusion can have beneficial effects on the maturation of hiPSC-derived liver spheroids when applied directly at the embryoid body stage of differentiation, including the increased expression of albumin, CYP3A4 and FOXA2 [25] and higher albumin secretion [25,26]. These findings are encouraging and prompt the approach that MPS should be used in the differentiation process and not just used as a tool to culture the finalized models. A recent study investigated the effect of a dynamic microfluidic system on the sensitivity of a primary hepatocyte-based model to detect hepatotoxicity [27]. The authors showed that a dynamic culture could maintain hepatocyte functionally longer than conservative culture models. The current state of knowledge indicates that the perfusion of liver models in MPS can improve the maintenance of liver models and can improve differentiation from hiPSC when applied during the early stages of the differentiation. 

Similar to the liver spheroids, the BMEC-like cells maintained their status in the chip. The TEER values were not as high as described in other publications using iPSC-derived BMEC-like cells [28,29]. Although, these protocols use membrane models with a bigger surface area such as 12-well cell culture inserts. It was previously shown that TEER values in vitro are dependent on the surface area and circumference of the used cell culture insert, resulting in an underestimation of the real TEER value in smaller culture inserts [30]. A reduction in TEER values after an initial peak over the following week was observed in these protocols as well. Interestingly, we observed that the cell density of the BMEC-like cells on the membrane increased, while cell and nuclei size decreased over time. A higher cell density also means a higher total length of cell–cell junctions, which directly correlates to the area available for paracellular transport. Therefore, increasing cell numbers on the membrane could be related to the decreased TEER values after seven days of culture. 

While effects of shear stress on primary BMECs are described to be overall beneficial for cell functionality [31], in hiPSC-derived BMEC-like, the effect is not as pronounced. Two independent studies [7,32] found that shear stress did not alter or improve the barrier functionality of hiPSC-derived BMEC-like cells tremendously. While DeStefano et al. found no differences in the expression levels of key BBB markers such as PGP, VE-cadherin, claudin-5, occludin, GLUT1 and ZO-1, Vatine et al. detected some changes in the gene expression of members from the claudin family and general endothelial markers such as PECAM1 and vWF. They also described shear stress dependent changes in cholesterol biosynthesis, smooth muscle contraction, cell migration and angiogenesis by gene ontology analysis of mRNA sequencing data. In the Chip4, the average shear stress in the brain compartment can be estimated to be 0.082 dyne/cm² (see Appendix A). Although this is about two orders of magnitude lower than reported in vivo shear stress values of 5–23 dyne/cm² in the capillaries [33], the effects of fluid flow have been reported already at an average shear stress of only 0.01 dyne/cm² [7].

Differences in transcript expression before and after chip culture have been observed for multiple tight-junction-associated genes, genes associated with water transport and genes associated with the cytoskeleton. Of the differentially expressed genes only cytokeratin 18 and MUC18 downregulation were associated directly with the chip culture, while other changes were rather dependent on additional factors such as coculture with neural spheroids, additional culture time and the coculture medium. Interestingly, it has been described that during embryonal development, MUC18 expression is downregulated upon pericyte recruitment and is accompanied by the maturation and tightening of the newly formed BBB [34]. This is the first indicator that shear stress could have a positive effect on BMEC maturation in vitro.

The downregulation of claudin−1 and the upregulation of claudin-11 have been described upon hypoxic conditions in a mouse model [35]. The opposite regulation observed here might indicate that the cells experience less hypoxic stress once the cells are settled on the membranes. Aquaporin−3 is described to be one of the most prominent aquaporins in healthy endothelium facilitating water and glycerol effluxes [36]. Claudin-8 is not a typical claudin of the BBB, but is described to be involved in fluid resorption in airway epithelial cells [37], which in context with the increased aquaporin-3 expression points towards an increased need for fluid exchange, potentially caused by the integration of the brain model.

Overall, like the study of DeStefano et al. [32] we have found no major changes of iPSC-derived BMEC upon shear stress exposure. Additionally, as other groups before, we observed no morphological changes such as elongation [7,32,38]. This is a special morphological feature of BMEC-like cells not seen in other endothelial cells, which is hypothesized to have developed to minimize the length of cell–cell junctions over the vessel length [39].

The introduction of BMEC-like cells led to the formation of nutrient and macromolecules gradients between the brain compartment and the medium circulation. While albumin and LDH gradients showed the functionality of the formed barrier, the accumulation of lactate was not expected. Lactate concentration in the brain compartment was significantly higher with BMEC-like cells, indicating that the BMEC-like cells hindered lactate clearance from the brain compartment. In vivo*,* lactate is found at only minor concentrations in the brain under physiological conditions, where it acts as a signaling molecule or can be used by neurons as an energy supply instead of being a metabolic end-product [40]. In vivo, the BBB allows lactate equilibration between brain tissue and blood through monocarboxylate transporters such as MCT1 [41]. Insufficient lactate equilibration indicates that the exchange area between the two compartments was either not sufficient to allow the necessary exchange rate or that monocarboxylate transporters were not functional. The further increase upon BMEC-like cell addition would not be expected if the scaling of the brain tissue and the BBB area was homogenous. The brain model was scaled with a factor of 1:10^5^ from an approximated number of 1 × 10^11^ neurons in the human brain to 1 × 10^6^ cells to the in vitro model. The model of the BBB, due to the given size of the Chip4, is scaled by the factor 1:10^6^ from a BBB surface area of 12 to 18 m² in vivo [42] to 0.14 cm² in the in vitro model. Therefore, it is likely that the BBB model at this size is not able to provide sufficient area to allow lactate equilibration between the compartments. A potential solution for this imbalance could be to reduce the number of cells in the brain compartment to match the scaling factor that is given for the BBB model to reduce the lactate load. As described before, the cell number was scaled down from the total number of the brain. Arguably, the used neural spheroids are not ideally suited to model the whole brain as an organ and rather mimic the cerebral cortex. If following the logic of modelling the cerebral cortex instead of the brain in general, lower cell numbers would also be indicated from the perspective of scaling. 

The permeation behavior of atenolol and propranolol in the Chip4 system matches their BBB permeation properties in vivo. This distribution behavior of the two substances was only observed when BMEC-like cells were present in the system, highlighting their importance in modelling the BBB. In vivo, propranolol concentration in the brain tissue exceed the plasma concentration ~26-fold [43], compared to an only slightly but significant increased ratio in the Chip4. The pronounced difference in vivo is created through the rapid metabolization and excretion of the drug. Since the metabolization capability of the hiPSC-liver model is rather low and no renal model was present, this difference between in vivo and in vitro behavior can be expected. In vivo*,* propranolol concentrations after a single oral administration of 160 mg of propranolol are described to peak at 2 h with concentrations of 202.2 to 245.0 ng/mL and to decline to 10.2 to 19.4 ng/mL at 24 h [44]. In the Chip4, propranolol concentrations were still at 880 ng/mL 48 h after application. Another factor worth considering is that in vivo propranolol is highly bound by the brain tissue due to its high lipophilicity, with a brain/CSF ratio of 273 [43]. In this setup, the brain model binds only 4.94% (±5.16% SD) of the permeated propranolol as measured in a static Transwell setup (data not shown). Propranolol is also a strongly protein-bound substance, which in vivo is bound around 90% by plasma proteins. In vivo, the unbound propranolol in the plasma equilibrates with both the bound drug in the plasma and the free drug in the brain [43]. Since in the brain compartment the same culture medium is used as in the rest of the chip system, equilibration of bound and unbound substance is expected to behave similarly in the brain compartment.

The mean brain/circulation ratio of atenolol was found to be 0.28, which is similar than the reported in vivo ratio of 0.2 [43]. In vivo, atenolol plasma concentrations peak after a single oral application of 100 mg at 600 ng/mL, 3 h after application [45], whereas in the Chip4, even after 48 h 1900 ng/mL, atenolol is found in the circulation, which also can be attributed to the missing excretory capability of the setup. This is one of the current limitations of the system, which can be overcome in the future by the integration of kidney models that allow substance excretion.

Cell-specific metabolization was observed for both compounds in the Chip4 with the phase II glucuronidation of propranolol by the liver model and of atenolol by the neural model. The oxidation of propranolol to 4-hydroxy propranolol is the major route of metabolism in vivo, but the reaction was missing in the Chip4. Nevertheless, the missing formation of 4-hydroxy propranolol could also be associated with the missing intestinal models, as it was shown that the metabolite is only formed upon oral but not intravenous application [46]. Besides oxidation, glucuronidation is a major pathway of propranolol metabolism in vivo [47]. Propranolol-glucuronide was formed in the Chip4 and was only minimally able to pass the BMEC-like cell barrier. This could be expected, because it is known from other substances such as morphine that glucuronidation can drastically reduce BBB permeability [48].

In vivo*,* atenolol is metabolized by hydroxylation and glucuronidation [49]. Glucuronidated but no hydroxylated atenolol was found in the Chip4. Instead, desaturated as well as dealkylated atenolol was found. The desaturated form is likely to be formed spontaneously, since this metabolite was also found in empty control chips. The dealkylation of atenolol is not described to occur in vivo, but is described for other beta-blockers such as propranolol and metoprolol [50,51].

With the current state of iPSC-derived liver models, only incomplete maturation is achieved, therefore also the metabolization of compounds does not yet faithfully depict the situation in vivo*,* which is still a limitation of the here-presented system. As an intermediate solution, the BBB permeation of metabolites could be studied by the direct addition of the major known metabolites. Although, as soon as iPSC-derived liver models are able to truly model donor-specific metabolization, such an MPS as presented here will be superior to this approach. The on-chip metabolization allows the modelling of dynamic donor-specific metabolite formation and distribution, which is not possible if metabolites are directly added.

For following studies, the whole Chip4 system including autologous intestinal and kidney models to enable enhanced pharmacokinetic and pharmacodynamics studies in these systems will be used. Ultimately these systems are thought to develop further into an universal physiological template that we recently termed organismoid when derived from a single donor [52]. Furthermore, the model of the neurovascular unit can be improved by adding additional layers of complexity in the future. These could include addition of secondary cell types such as pericytes and microglia and the replacement of PET membranes by biological membranes. Finally, a recent transcriptomic meta-analysis of hiPSC-derived BMEC-like cells demonstrated an epithelial fingerprint in these cells, raising doubt regarding their pure endothelial genetic background [53]. Missing endothelial lineage genes might affect the potential of these cells to recapitulate the BBB in specific applications. Therefore, other protocols yielding a more classical endothelial cell phenotype [19] should be explored for their ability to model the BBB.

## Figures and Tables

**Figure 1 cells-11-03295-f001:**
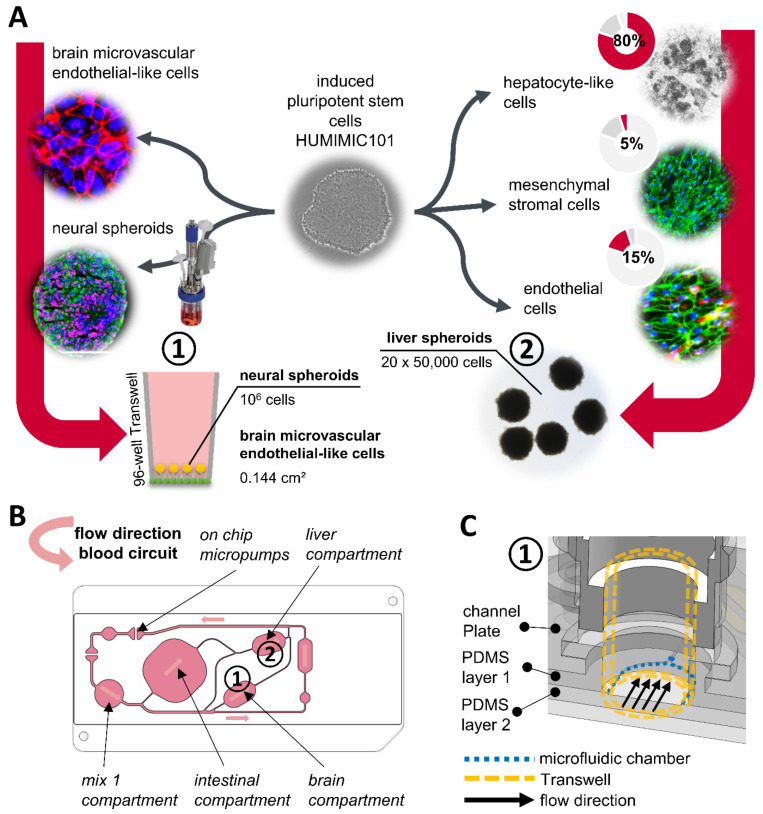
Setup of the iPSC-derived BBB/brain and liver coculture assay in the HUMIMIC Chip4. (**A**) Schematic iPSC differentiation of the HUMUMIC101 iPSC line into brain microvascular endothelial-like cells, neural spheroids, endothelial cells, mesenchymal stromal cells and hepatocyte-like cells. Brain microvascular endothelial-like cells and neural spheroids were combined in 96-well Transwells to build a blood–brain-barrier / brain model ①. The endothelial cells, mesenchymal stromal cells and hepatocyte-like cells were combined in spheroids of 50,000 cells, and 20 of those spheroids per chip were used as a liver model ②. (**B**) 2D view of the HUMIMIC Chip4 microfluidics; the surrogate blood circuit is shown in pink. In the depicted circuit a medium reservoir (mix 1) is interconnected with a 24-well intestinal compartment and 96-well compartments for the liver ① and BBB/brain ② model. The fluid flow is created by on-chip micropumps; the direction of the flow is indicated by arrows. (**C**) 3D view of the brain culture compartment ① in the HUMIMIC Chip4. The bottom of the compartment consists of the PDMS layer 2. At the sides, the compartment consists of the PDMS layer 1 and the channel plate. Cut-off 96-well Transwells (yellow dotted line) can be inserted into the compartment and stand at their edges on a 100 µm high step of PDMS layer 2. Endothelial cells cultured at the bottom of the Transwell membrane are thereby directly exposed to the fluid flow passing underneath.

**Figure 2 cells-11-03295-f002:**
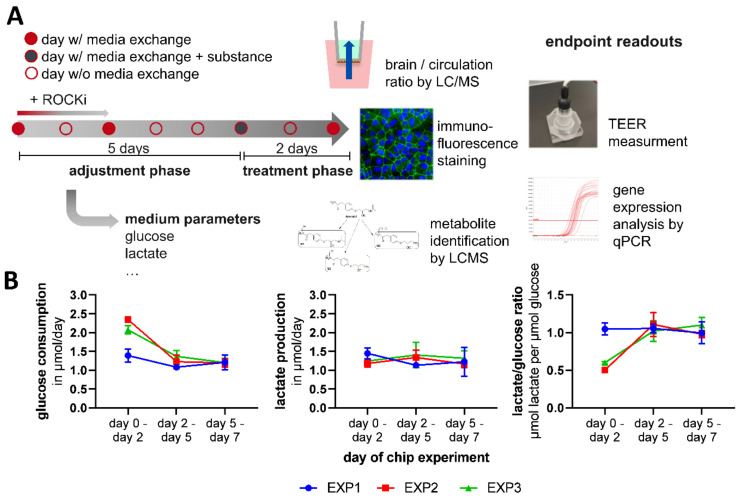
Cultivation schedule and medium parameters of the one-week coculture. (**A**) Schematic of the dynamic coculture in the HUMIMIC Chip4, with in-process measurement of medium parameters and end-point analysis. Organ models are cultured for 7 days in total in the Chip4, with a 5-day adjustment phase and a 2-day treatment phase. Medium is exchanged every 2–3 days. At the end of the assay, different endpoint readouts such as the measurement of substance concentration by LC/MS, TEER measurement, immunofluorescence staining, metabolite identification by LCMS and gene expression analysis are performed. (**B**) Glucose consumption, lactate production and lactate/glucose ratio from three experiments (N = 3). Mean values + s.e.m. per experiment with 3 to 6 technical replicates (*n* = 3–6) per timepoint are shown.

**Figure 3 cells-11-03295-f003:**
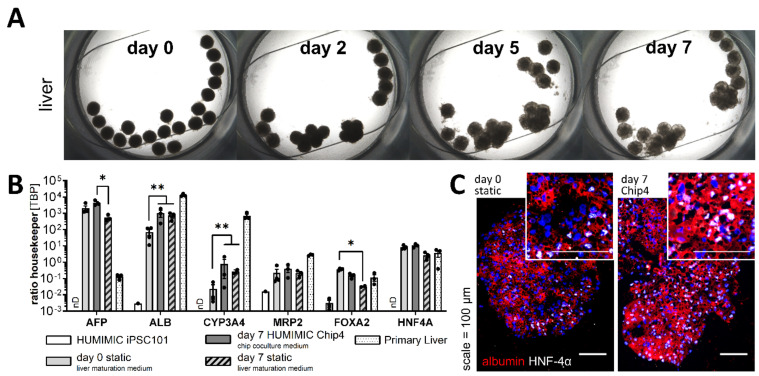
Liver models in the HUMIMIC Chip4. (**A**) Brightfield images of the liver compartment with 20 liver spheroids over one week in the HUMIMIC Chip4 in coculture with the BBB/brain model. (**B**) Transcript levels of liver-associated genes in iPSC-derived liver spheroids after formation (day 0) and after 7 days of Chip4 culture in the chip coculture medium and after 7 days of static culture in liver maturation medium. Data are expressed as mean ± s.e.m of three independent experiments (N = 3). On the *y*-axis, the gene expression levels are shown as ratio to the housekeeper gene TBP. HUMIMIC hiPSC101 samples from three different passages and primary liver samples from three donors were used as controls. Two-way ANOVA with Tukey post hoc test, ** *p* < 0.01, * *p* < 0.05. (**C**) Immunofluorescence staining of liver markers albumin in red and HNF-4α in white in iPSC-derived liver spheroids before and after 7-day culture in the Chip4; scale is 100 µm. Cell nuclei were stained with DAPI in blue.

**Figure 4 cells-11-03295-f004:**
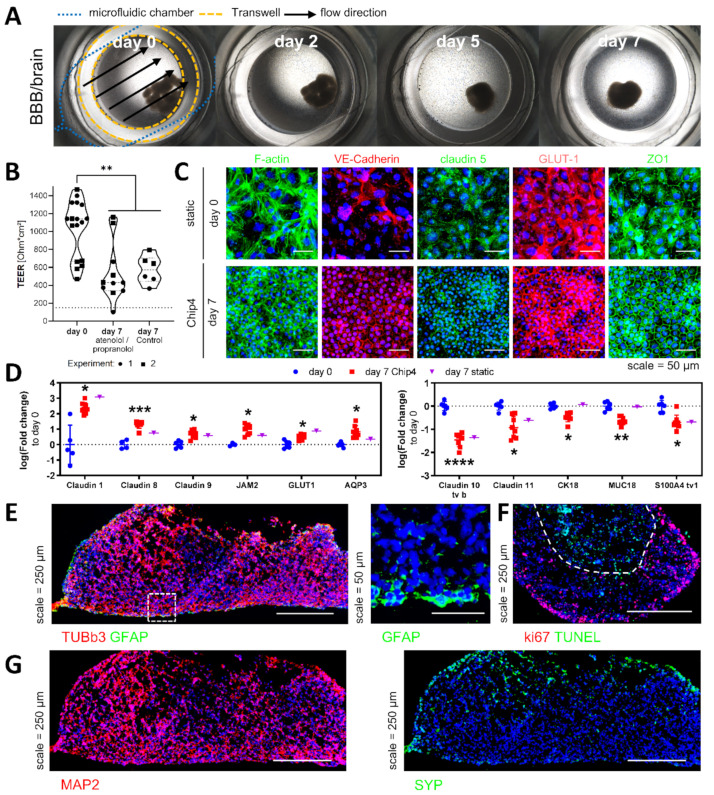
Organ models in the HUMIMIC Chip4: liver and BBB/brain model. (**A**) Brightfield images of the BBB/brain compartment over one week in the HUMIMIC Chip4 in coculture with the liver model. In the image of day 0 of culture, exemplarily, the location of the microfluidic chamber, of the Transwell and the flow direction are shown. (**B**) TEER values before and after 7-day culture in the Chip4 with and without substance addition, from two experiments (N = 2) with 3–5 chip replicates per group (*n* = 3–5). One-way ANOVA with Tukey post hoc test, ** *p* < 0.01. (**C**) Immunofluorescence staining of endothelial and BBB marker in iPSC-derived BMEC-like cells culture before and after 7-day culture in the Chip4; scale is 50 µm. (**D**) Significantly upregulated and downregulated genes in BMEC-like cells after chip culture compared to day 0 and to day 7 in static culture. Data are expressed as a logarithmic fold change to the day 0 samples. Six samples from three experiments on day 0 (N = 3), eight samples after Chip4 cultivation in three experiments (N = 3) and one sample from day 7 in static culture (N = 1) were analyzed. The expression levels of individual samples and mean ± s.e.m are shown. Statistical significance was determined by multiple *t*-tests using the Holm–Sidak method to correct for multiple comparisons, **** *p* < 0.0001, *** *p* < 0.001, ** *p* < 0.01, * *p* < 0.05. Each gene was analyzed individually, without assuming a consistent standard deviation (SD). (**E**) Left image: immunofluorescence staining of TUBb3 positive neuronal progenitor and GFAP positive glial progenitors in neural spheroids after 7-day culture in the Chip4; scale is 250 µm. Right image: Magnification of the GFAP staining in the region marked in the left image; scale is 50 µm. (**F**) Immunofluorescence staining of ki67-positive proliferating cells and TUNEL-positive apoptotic in neural spheroids after 7-day culture in the Chip4; scale is 250 µm. Border between the outer layer of proliferating cells and apoptotic cells in the spheroid core is marked by a dashed line. (**G**) Immunofluorescence staining of MAP2 positive neurons and synaptophysin (SYP) as a marker of synapse formation, stained after 7-day culture in the Chip4; scale is 250 µm. For all immunofluorescence staining: cell nuclei were stained with DAPI in blue.

**Figure 5 cells-11-03295-f005:**
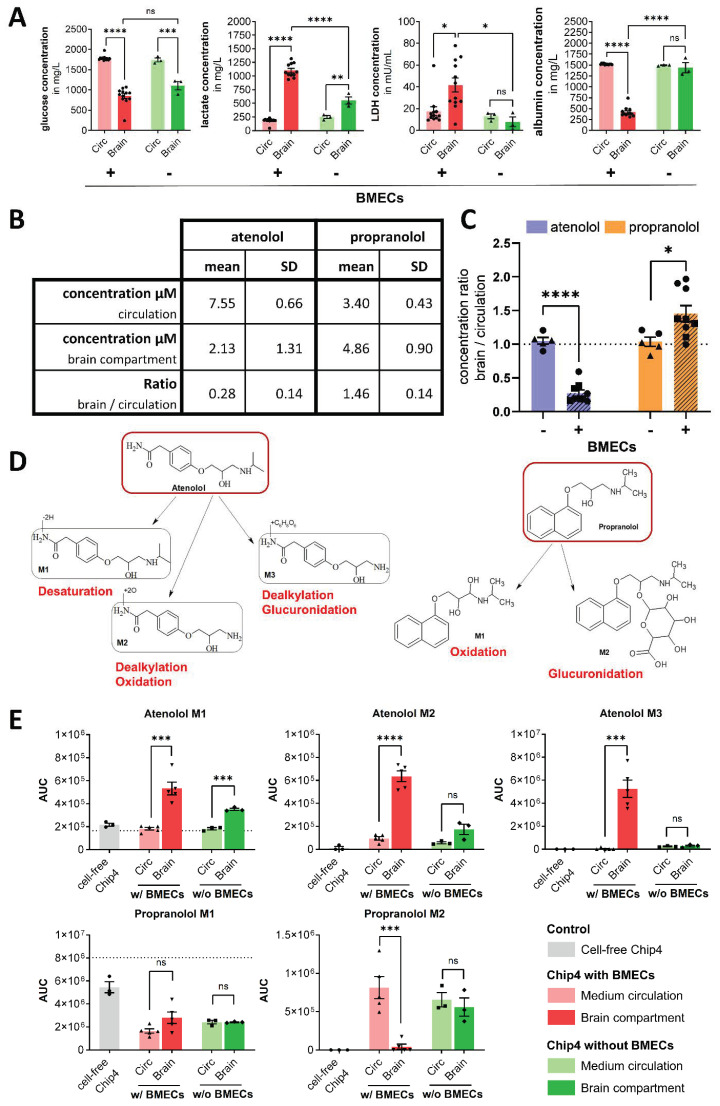
Permeation and metabolization of atenolol and propranolol in the Chip4. (**A**) Concentration of glucose, lactate, LDH and albumin in the circulation (Circ) and the brain compartment after 5 days of Chip4 culture. Twelve chips with BMEC-like cells (*n* = 12) and three chips without BMEC-like cells (*n* = 3) in the presence of neural and liver spheroids from one experiment were analyzed. On the *y*-axis, substance concentrations of individual circuits and mean values + s.e.m. are shown. Concentration differences were compared by two-way ANOVA with Tukey post hoc test (**** *p* < 0.0001, *** *p* < 0.001, ** *p* < 0,01, * *p* < 0.05). (**B**) Measured concentrations and ratios between the brain compartment and the medium circulation of propranolol and atenolol in the Chip4, 48 h after substance application. Mean values of two experiments (N = 2) with four and five chip replicates each and standard deviation (SD) are shown. (**C**) Measured concentration ratio between brain compartment and medium circulation of propranolol and atenolol, with and without BMEC-like cells from two experiments (N = 2) are shown with three and five chip replicates per condition (*n* = 3–5). On the *y*-axis, mean concentration ratios + s.e.m. are shown. Differences between circulation and brain compartment were compared by an unpaired *t*-test (* *p* < 0.05, **** *p* < 0.0001). (**D**) Proposed metabolic pathway for propranolol and atenolol in the Chip4. An identical instrument response is assumed for parent and metabolites. Identification of atenolol metabolites is based on accurate mass only. (**E**) Distribution of propranolol and atenolol metabolites between medium circulation (Circ) and the brain compartment 48 h after substance application. Three cell-free Chip4 (*n* = 3), five Chip4 with BMEC-like cells, neural and liver spheroids (*n* = 5) and three Chip4 without BMEC-like cells and with neural and liver spheroids (*n* = 3) were analyzed. On the *y*-axis, the area under curve (AUC) values of individual circuits and mean values + s.e.m. are shown. The dotted line shows concentration in the application medium. Differences between circulation and brain compartment were compared by unpaired *t*-tests (**** *p* < 0.0001, *** *p* < 0.001).

## Data Availability

The datasets generated during and/or analyzed during the current study are available from the corresponding author on reasonable request.

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
