# Peer review of "A Human Stem Cell-Derived Brain-Liver Chip for Assessing Blood-Brain-Barrier Permeation of Pharmaceutical Drugs"

_cells, 2022, doi:10.3390/cells11203295_

Round 1

Reviewer 1 Report

The topic of the manuscript entitled "A human stem cell-derived brain-liver chip for assessing blood-brain-barrier permeation of pharmaceutical drugs" is timely and the paper is overall well presented; there seems to be some missing information/mistake/missing legend in Fig 5 with regard to color coding in the graphs.

With regard to the discussion, the sentences starting in line 421 "A potential solution for this issue, could be to only model the cerebral cortex instead of the whole brain. This would require lower numbers of neural cells, which would reduce the lactate load in the brain compartment." are difficult to grasp in context of the paper; how could the transwell-cultured layer of neurons resemble "whole brain"; and is the number of cells in a compartment not just a resultant of the seeding density? Probably, there is an explanation as such that these statements make sense, however it does not follow directly from the context of the manuscript. Please improve the discussion section accordingly to leave no misunderstanding over what the model can/cannot offer and what you actually demonstrate by experiment.

Author Response

Point 1: There seems to be some missing information/mistake/missing legend in Fig 5 with regard to color coding in the graphs.

The color coding in this graph is not giving additional information to the labels of the graphs and was only meant to give additional visual guidance for the reader. We agree that this might be confusing so we added a color legend in the bottom right corner of the figure to guide the reader. Also, we changed the color code of the C graph from grey to blue/orange to avoid confusion with the grey bars in figure E, which are the empty chip controls.

Point 2: With regard to the discussion, the sentences starting in line 421 "A potential solution for this issue, could be to only model the cerebral cortex instead of the whole brain. This would require lower numbers of neural cells, which would reduce the lactate load in the brain compartment." are difficult to grasp in context of the paper; how could the transwell-cultured layer of neurons resemble "whole brain"; and is the number of cells in a compartment not just a resultant of the seeding density? Probably, there is an explanation as such that these statements make sense, however it does not follow directly from the context of the manuscript. Please improve the discussion section accordingly to leave no misunderstanding over what the model can/cannot offer and what you actually demonstrate by experiment.

We simplified this part of the discussion which can now be found in the lines 607 – 614. As you pointed out the bottom-line of the suggested improvement is simply a reduction of seeding numbers in the brain compartment. Inside the Transwell not a monolayer of neurons but neural spheroids are cultured. However, we agree that even the neural spheroids do not mimic the whole brain but rather the cerebral cortex. Modelling only the cerebral cortex, from a scaling point of view would also require lower cell numbers than if the approach was to mimic the whole brain. In that paragraph we tried to point out that the reduction of cells numbers would make sense both from the metabolical as well as a physiological perspective. We think that with the rephrased paragraph this is now better explained. We also included two additional notes (lines 643-645, 686) on the current limitations of the system to leave no misunderstanding over the current capabilities of the assay.

Reviewer 2 Report

The submitted manuscript describes high quality science. The main text is accompanied with well organized supplemental figures, experimental procedures and additional references.

The reviewer suggest the authors to use the "Cells" template to adopt the right format for the journal.

Part of the experimental procedures should be included in the main text to help readers.

Figure 1 A is difficult to understand and need more explanations

The legends to some figures (e.g. figure 1, what is B and C ?) should be improved to include information so the the figure could be understood without referring to the main text.

Author Response

Point 1: The reviewer suggest the authors to use the "Cells" template to adopt the right format for the journal.

The article was not adopted to the MDPI format following the comments of the editor.

Point 2: Part of the experimental procedures should be included in the main text to help readers.

We moved the experimental parts regarding the cell culture and the statistical analysis to the main text to make this information more easily available to the reader.

Point 3: Figure 1 A is difficult to understand and need more explanations. The legends to some figures (e.g. figure 1, what is B and C ?) should be improved to include information so the figure could be understood without referring to the main text.

We noticed that for Figure 1, part of the description was lost during the submission of the article. We included that part again into the figure legend and extended the description of the figure in general to allow the reader to understand the figure without referring to the main text. We also updated the description of Figure 2 to better guide the reader through the process.